# Sulfuric Acid Baking—Water Leaching for Gold Enrichment and Arsenic Removal from Gold Concentrate

**Bongju Kim [1,†], Chulhyun Park [2,†], Kanghee Cho [3], Jaehyun Kim [4], Nagchoul Choi [2,\*] and Soonjae Lee [4,\*]**

1   Radioactive Waste Disposal Research Division, Korea Atomic Energy Research Institute, Daejeon 34057, Korea; kbj7878@kaeri.re.kr
2   Department of Advanced Energy Engineering, Chosun University, Gwangju 61452, Korea; chpark@chosun.ac.kr
3   Department of Rural Systems Engineering, Research Institute of Agriculture and Life Sciences, Seoul National University, Seoul 08826, Korea; kanghee1226@snu.ac.kr
4   Department of Earth and Environmental Sciences, Korea University, Seoul 02841, Korea; jhyun701@korea.ac.kr
\*   Correspondence: nagchoul@snu.ac.kr (N.C.); soonjam@korea.ac.kr (S.L.); Tel.: +82-62-230-7878 (N.C.); +82-2-3290-3177 (S.L.)
†   These authors contributed equally.

**Abstract:** During the roasting of gold concentrate to improve gold recovery, arsenic is released into the air and valuable elements such as Fe, Cu, Zn, and Pb are converted into oxide minerals. In this research, we evaluated the release of As and the loss of valuable metals during the acid baking and hot water leaching processes used for gold concentrate. The acid bake tests were conducted for gold concentrate using an electric furnace by applying various concentrations of $H_2SO_4$ solution under different baking times. The water leaching process was enacted using 70 °C water for the baked samples. Chemical and mineral compositions of the raw and treated samples were analyzed using AAS and XRD, respectively. The results show that soluble metal sulfates, such as rhomboclase and mikasite, were formed in the baked samples, and that the leaching of valuable metals (Fe, Cu, Zn, and Pb) was accelerated during the hot water leaching procedure. During acid baking, arsenic was partially removed by volatilization, and the rest of the arsenic-containing minerals were converted to soluble minerals. The soluble arsenic-containing mineral resulted in a dissolution that was 60 times higher than in the roasted sample. The maximum gold grade of solid residues increased up to 33% through the acid baking–water leaching process. It was confirmed that acid baking with $H_2SO_4$ prevented As release into the air, as well as the recovery of valuable metals through hot water leaching, such as Fe, Cu, Zn, and Pb, which were formerly discarded in the tailings.

**Keywords:** gold concentrate; acid baking; soluble metal sulfate; hot water leaching; arsenic

## 1. Introduction

Gold is mainly contained in pyrite ($FeS_2$) and arsenopyrite (FeAsS). Since gold contained in pyrite and arsenopyrite is small, smaller than a few microns, it is difficult to observe it with an optical microscope or even with an electron microscope. This gold is called invisible gold, and it is notoriously difficult to liberate it from sulfide mineral. The reason for this difficulty is because invisible gold typically forms a solid solution with sulfide minerals or is chemically bound with sulfides [1]. To overcome this problem, the roasting of pyrite or arsenopyrite containing invisible gold, at a high temperature, is a commonly used method [1,2].

To select a treatment process for gold enrichment, the behavior of other precious metals and impurities, as well as the effect of gold enrichment, should be considered. In addition, the environmental problems that may occur during the treatment process are also important considerations. Pyrite and arsenopyrite become pyrrhotite and hematite after roasting due to phase transformation, and a series of oxidation, volatilization, and

removal is performed as S in sulfide mineral changes to $SO_2$, and As changes to $As_2O_3$. As a result of roasting sulfide mineral, gold leaching is improved due to enhanced leaching through the pore structure developed by the evacuation of the volatile elements, such as S and As. As S and As are removed, a dense rim structure is formed on the surface of the mineral particle and porous hematite ($Fe_2O_3$) is formed inside; however, no phase change occurs in the core where the sulfide mineral is distributed [3]. Since the mineral surface is surrounded by a dense medium, gold solvent of an aqueous solution cannot be penetrated easily. Furthermore, $SO_2$ and $As_2O_3$ generated in the roasting process cause air pollution. $As_2O_3$ generated during the roasting process reacts with $Fe_2O_3$ hematite to form $FeAsO_4$. $FeAsO_4$ plays a role in inhibiting gold leaching [4,5]. Thus, it is challenging to achieve dead roasting where S and As are completely removed in the air [6].

As a process for the complete removal of arsenic, two-step roasting has been conducted [2,5,7]. In the first roasting step under the $N_2$ condition, arsenic was completely removed from gold-containing arsenopyrite and pyrite. In the second step, roasting under an aerobic condition, the iron minerals were transformed to porous hematite by the complete removal of S, so that the gold contained in porous hematite could be leached and extracted effectively with a cyanide or non-cyanide solvent. However, considerable costs and appropriate facilities are required to implement the second step of the roasting.

Pyrite and arsenopyrite, in which invisible gold is commonly found, are produced along with chalcopyrite ($CuFeS_2$), covellite ($CuS$), chalcocite ($Cu_2S$), sphalerite ($ZnS$), pyrrhotite ($Fe_{1-x}S$), galena ($PbS$), and tetrahedrite ($Cu_{12}Sb_4S_{13}$). The complex sulfide minerals contain valuable metals such as Cu, Zn, Pb, and Fe, as well as impurities such as As, Bi, and Sb. In particular, gold is produced from the epithermal deposit, while elements such as Te, Bi and Sb are always produced from invisible gold ores. Thus, elements such as As, Bi, and Sb that are contained in gold concentrates play a role as impurities in ironworks [8–11]. Furthermore, these impurities are emitted into the atmosphere, resulting in environmental pollution during the roasting process.

Although gold is leached and extracted using a cyanide or non-cyanide solvent, with regard to roasting minerals, valuable metals such as Fe, Cu, Zn, and Pb, which are contained in roasting minerals, are regarded as tailings [12]. The existence and value of these valuable metals have almost been ignored in the gold leaching process. During the roasting process, the valuable metals contained in gold concentrate are transformed into oxides, thereby reducing the efficiency of separation and recovery. Thus, it is necessary to establish an eco-friendly pre-processing technique that can improve gold grade but also avoid air pollution while leaching these valuable metals and impurities contained in gold concentrates effectively.

In recent years, studies have been conducted on valuable metals such as arsenopyrite, enargite ($Cu_3AsS_4$), tennantite ($Cu_{12}As_4S_{13}$), molybdenite ($MoS_2$), low grade nickel ores, spent catalyst, and nickel-containing laterite using acid baking-water leaching. The acid baking-water leaching process refers to a method of leaching valuable metals such as Cu [13], Fe [14], Zn [15], As [13], Co, Ni, Mn, Al [12], Mg, and Cr by mixing sulfide minerals with a concentrated sulfuric acid solution followed by heating, i.e., baking them in an electric furnace at a relatively low temperature, the range of which is 300 °C to 500 °C, for about four hours or longer, and then dissolving the baking products with 70 °C water [14–21].

The roasting process refers to a method that heats sulfide minerals at a very high temperature, over 700 °C, thereby oxidizing all sulfide minerals. Thus, acid baking–water leaching saves energy because heating is carried out at a relatively low temperature, approximately 500 °C or below. Once the sulfide minerals are roasted, air pollutants such as $SO_2$ and $As_2O_3$ are produced. However, air pollutants of $SO_2$ are not emitted in acid baking-water leaching. Rather, this process acts as sulfation, thereby contributing to metal leaching, while $As_2O_3$ is leached and extracted in water leaching. Since Fe contained in chalcopyrite ($CuFeS_2$), pyrite ($FeS_2$) and arsenopyrite ($FeAsS$) is transformed into hematite ($Fe_2O_3$) or magnetite ($Fe_3O_4$), the removal of Fe is difficult. When these Fe-containing

sulfide minerals are acid baked, they are transformed to $FeSO_4$ or $Fe_2(SO_4)_3$ so that they can be easily dissolved and extracted with water leaching [22–24].

If sphalerite (ZnS) is roasted, insoluble ZnO is produced, which makes it difficult to leach and extract Zn. However, if acid baking is performed, $ZnSO_4$ is formed so that Zn can be easily leached and extracted with water [18]. If gold-containing invisible/resistant sulfide minerals are roasted at a high temperature, gold leaching is improved. Here, impurities such as As and Bi are volatilized and emitted into the atmosphere. Furthermore, since Cu, Zn, Mn, Co, Ni, and Fe are transformed into oxides due to roasting, these are very difficult to leach and extract.

However, gold-containing invisible/resistant sulfide minerals are acid baked, and valuable metals as well as impurities can be leached and extracted with water leaching. In addition, since gold is not easily dissolved in a sulfuric acid solution, gold grade is improved in the solid residue. This study aimed to improve the gold grade of invisible gold concentrations in leaching valuable metals and to remove penalty elements using the acid baking-water leaching technique.

## 2. Materials and Methods

### 2.1. Sample

The gold concentrate sample was supplied from Golden Sun Co. Ltd. (Jeollanam-do, Korea). This sample was recovered after the flotation of gold-containing sulfide mineral concentrate [24]. The mineral composition and chemical composition of the gold concentrate were analyzed using X-ray powder diffraction (XRD) and atomic absorption spectrophotometry (AAS) after aqua regia digestion, respectively. In Figure 1, quartz, pyrites, covellite, and muscovite ($K(OHF_2)_2Al_3Si_3O_{10}$) are shown in terms of the XRD analysis. The chemical composition of the raw sample is presented in Table 1. The raw sample contained Au of 130.2 mg/kg, Ag of 986.6 mg/kg as well as the valuable metals, Fe of 86,793.1 mg/kg, Zn of 2709.6 mg/kg, and Cu of 1899.4 mg/kg. Additionally, the penalty elements As, Bi, and Pb were also found in significant levels (2456.1, 140.1, and 621.3, respectively). For the gold concentrate containing As, a penalty would be charged if As exceeded 0.2 wt.%

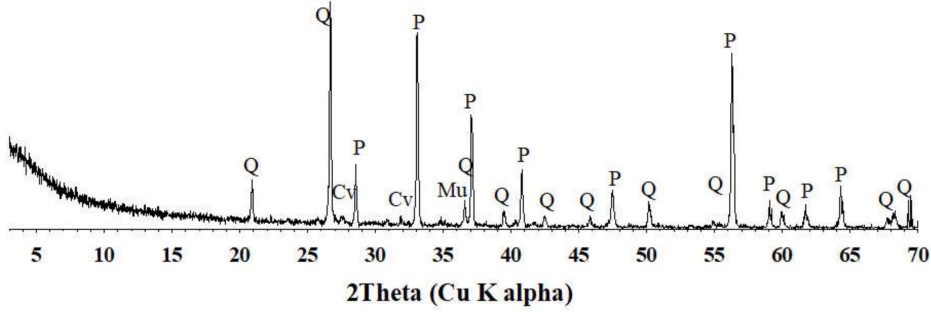

**Figure 1.** XRD pattern of gold raw concentrate. Cv, covellite; P, pyrite; Q, quartz; and Mu, muscovite.

**Table 1.** Chemical composition of gold concentrate from Golden Sun mine (unit: mg/kg).

| Elements | As | Cu | Fe | Zn | Pb | Bi | Au | Ag |
|---|---|---|---|---|---|---|---|---|
| Concentrate | 2456.1 | 1899.4 | 86,793.1 | 2709.6 | 621.3 | 140.1 | 130.2 | 986.6 |

### 2.2. Acid Baking-Water Leaching Experiment

2.2.1. Acid-Baking

Acid baking and roasting for the gold concentrate were conducted in an electric furnace (Ceber Company, ceramic fiber SF-series). The gold concentrate sample (10 g) was prepared in an alumina crucible (inner diameter = 4.4 cm, height = 3.2 cm). The acid baking was conducted by adding 5.0 mL (1.0 M) of $H_2SO_4$ solution to the gold concentrate in the alumina crucible, and heated. The heated samples were cooled down in

the ambient condition and their weight was measured. As a control, the roasted sample was prepared by heating the sample at different temperatures for three hours without $H_2SO_4$ solution. To compare the effectiveness of the temperature, the roasting and acid-baking experiments were conducted at different temperatures, from 100 °C to 500 °C. The effect of operation conditions on the efficiency of the acid-baking was tested by baking at 400 °C by applying different concentrations of $H_2SO_4$ and baking times. The mineral composition and chemical composition of the concentrate samples recovered from each experiment were analyzed using XRD and AAS, respectively. Figure 2. shows the flowsheet for the experimental procedure.

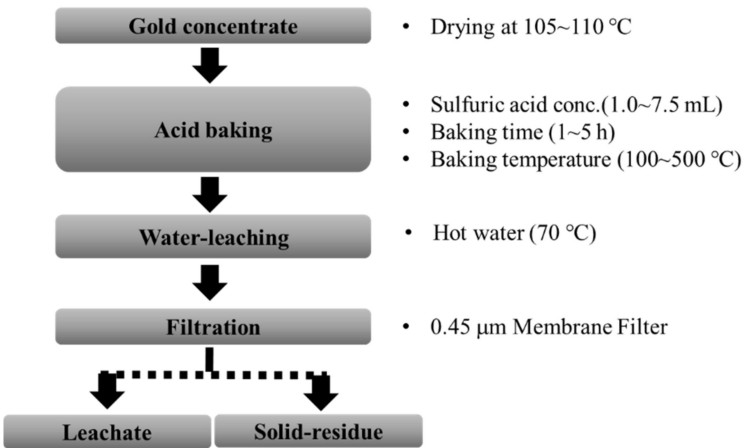

**Figure 2.** Flow sheet for the sulfuric acid baking-water leaching process.

### 2.2.2. Water Leaching

Water leaching was conducted for the baked samples. The baked sample (2.0 g) obtained for each experiment was put into an Erlenmeyer flask containing 100 mL of distilled water [14]. The preliminary tests showed that the amount of water added in the range 10 wt.% to 40 wt.% did not affect the leaching efficiency of metals; therefore, the water added was subsequently fixed at 50 wt.%. The Erlenmeyer flask was stirred at a rate of 250 rpm on a hot plate at 70 °C. While stirring, 5.0 mL of water-leaching solution was collected by time (10, 20, 30, 40, 50, and 60 min) from the Erlenmeyer flask. After completing water leaching, water-leaching solution and insoluble solid residue were filtered with 0.45 micron filter paper (hereafter referred to as water-leaching solution and solid residue). The solid residue was naturally dried followed by performing XRD analysis and aqua regia digestion. The contents of As, Fe, Cu, Zn, and Pb in the water leaching solution were analyzed by atomic absorption spectrophotometry (AAS). A leaching of the metal elements was expressed as a percentage of the content of metal element remaining in the solid residue from the content in the gold concentration. For As, a removal was used; for Au and Ag, an enrichment was used.

### 2.2.3. Aqua Regia Digestion

Part of the sample was treated by aqua regia digestion for the analysis of chemical composition. Each 1.0 g of the roasting and baking samples, and solid residue samples obtained from each experiment, was put into an 18 cm Pyrex test tube, and 4.0 mL of aqua regia was added to each sample. The test tube was heated for one hour at 70 °C in the heating block. After the aqua regia digestion solution was filtered with a syringe filter, As, Au, and Ag contents were measured with AAS (AA-7000, Shimadzu, Japan).

## 3. Results and Discussion

### 3.1. Effect of Acid Baking and Its Temperature on Grade Gold Concentrate

3.1.1. Removal of As and Enrichment of Au and Ag

Acid baking and roasting experiments were conducted at a temperature range of 100–500 °C. Figure 3 shows the change in chemical composition of gold concentrate under different temperature conditions of roasting and acid-baking. All the samples showed reductions in As content and increases in Au and Ag.

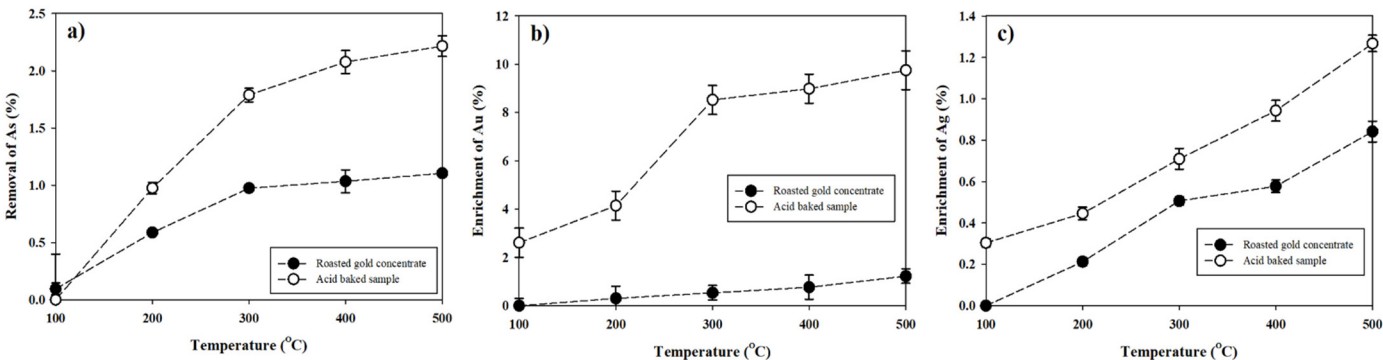

**Figure 3.** Effect of temperature on (**a**) removal of As, and (**b**,**c**) enrichment of Au and Ag.

In the roasting process (roasted for the samples being heated without acid [$H_2SO_4$]), as baking temperature increased, the loss of As in the roasting sample was increased up to 1.1% at 500 °C due to its volatility (Figure 3a). Arsenic can be volatilized and removed from gold concentrate, even by pyrolyzing during simple roasting [25]. In the acid-baking process using $H_2SO_4$, the As loss were about twice as high in the baking sample than those in the roasting sample; it increased up to 2.2% at 500 °C.

As a result of acid baking, the material flow was volatilized in small amounts in the case of As($As_2O_3$). In the acid baking experiment, arsenic exists in the form of arsenic acid in the solution, and in the case of Au and Ag, it exists in the form of AuO and $Ag_2O$ in the residue [12,26]. The acid-baked and roasted samples showed increases in Au and Ag (Figure 3b,c). In the case of Au, the enrichment was significantly increased from 1.2% at 500 °C (roasting sample) to 9.7% in the baked sample. The Ag content also increased as heat temperature increased. Even though the Ag-enrichment increased further upon acid baking, the difference was not dramatic as in Au.

3.1.2. Effect of Baking Temperature on Mineral Phase of Gold Concentrate

As the temperature of the acid baking process increased, the chemical composition of the recovered sample showed a great change, especially at 300 °C. This is related to the change in the mineral phase according to the temperature condition. The change in the mineral composition by acid baking was analyzed using XRD (Figure 4). The results show that quartz and pyrite were found in raw and baked samples. Covellite was found in the roasting sample. Muscovite was found in 100 °C baking sample. At 200 °C and 300 °C, rhomboclase (FeH($SO_4$)$_2$·$4H_2O$) and anglesite (PbSO$_4$) were found at diffraction values of 4.58 Å and 3.63 Å (rhombocase) and at 2.75 Å (anglesite), respectively [18]. At 300 °C, rhomboclase and anglesite were found with a diffraction value of 4.59 Å corresponded to rhomboclase, and diffraction values of 3.82 Å, 3.21 Å, and 2.75 Å corresponded to anglesite. At 400 °C, hematite (2.51 Å) was found. Additionally, in the 500 °C baking sample, hematite (3.67, 2.69, 2.51, 2.20, 2.08, 1.83, 1.69, 1.60, and 1.45 Å), magnetite ($Fe_3O_4$ = 2.94, 2.51, 2.08, 1.69, 1.60, and 1.47 Å) and anglesite (3.22 Å) were revealed. Although the d-values of hematite and magnetite overlapped, 3.67 Å and 2.69 Å implied hematite while 2.94 Å and 1.47 Å indicated magnetite.

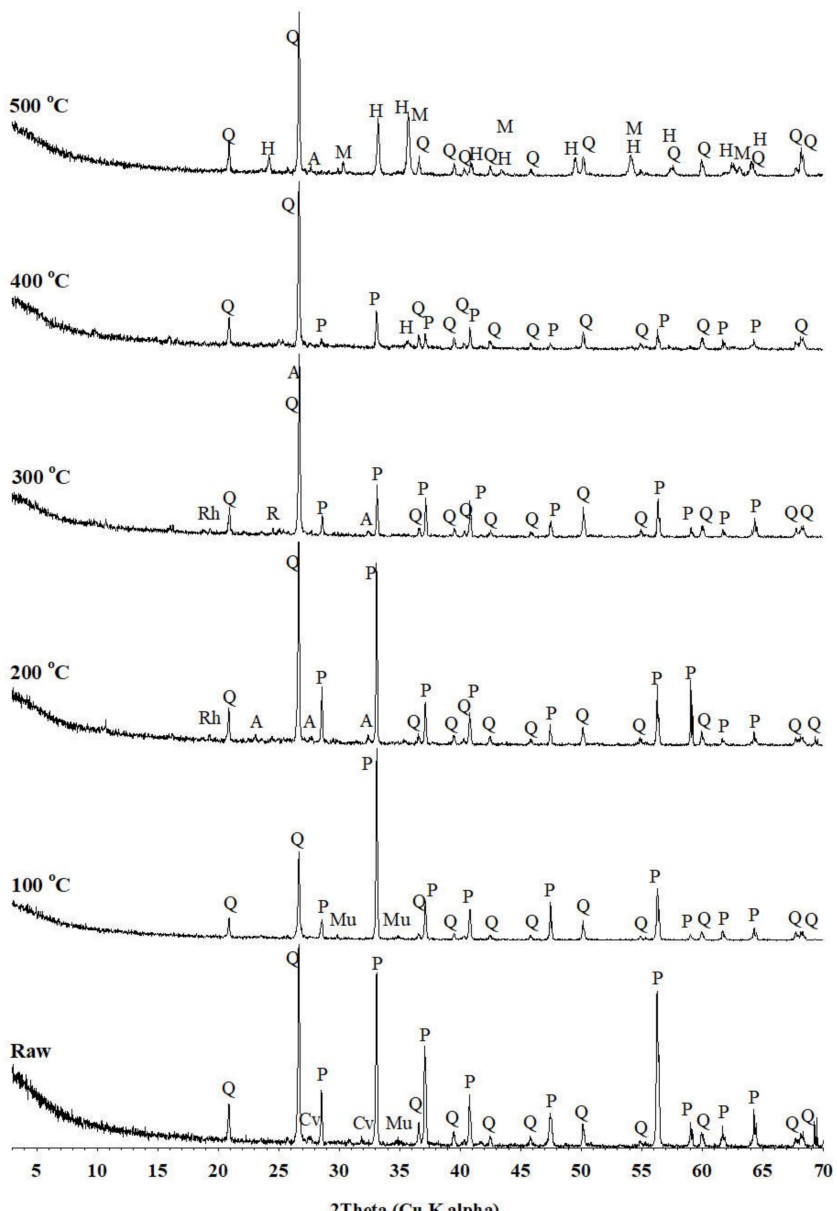

**Figure 4.** XRD patterns of raw and acid-baked samples in sulfuric acid baking for 3 h. A, anglesite; Cv, covellite; H, hematite; M, magnetite; Mu, muscovite; P, pyrite; Rh, rhomboclase; and Q, quartz.

Various mineral types and contents were observed to change according to the baking temperature. Rhomboclase (FeO) was exhibited at 200~300 °C, hematite (Fe$_2$O$_3$) at 400~500 °C, and magnetite (Fe$_3$O$_4$) at 500 °C. The soluble rhomboclase was pyrolyzed at 400~500 °C, and changed to hematite and magnetite. This suggests that the acid-baking temperature is related to mineral transformation, especially into terms of soluble metal sulfate. From the appearance of soluble minerals such as rhomboclase, acid baking with H$_2$SO$_4$ could be effective at 200~300 °C, since the insoluble valuable metals bonded with sulfur were transformed into soluble metal sulfates. Previous research also reported that acid baking using a sulfuric acid solvent could effectively recover valuable metals from sulfide minerals. The occurrence of soluble sulfate minerals can be described by Equations (1)–(4) [14].

$$CuS(Covellite) + H_2SO_4 = CuSO_4 + H_2S \qquad (1)$$

$$CuFeS_2(Chalcopyrite) + 2H_2So_4 = CuSO_4 + FeSO_4 + 2H_2S \qquad (2)$$

$$FeS_2(pyrite) + H_2SO_4 = FeSO_4 + S^o + H_2S \tag{3}$$

$$ZnS(sphalerite) + H_2SO_4 = ZnSO_4 + H_2S \tag{4}$$

### 3.2. Effect of Water Leaching on Grade Gold Concentrate

#### 3.2.1. As Removal

The water-leaching experiment was conducted for 10 min to 60 min with 2.0 g each of the roasting and acid-baking samples. The leaching of As is shown in Figure 5. In the water leaching process after acid baking, arsenic was leached with high efficiency. However, in contrast, As leaching from the roasted samples was insignificant. The sample baked at 400 °C showed a maximum As leaching of 78.20%. For the 500 °C baking sample, the leaching of As increased up to 74.92%. The lower As leaching from the roasted sample at 500 °C compared to 400 °C indicates that baking at temperatures higher than 400 °C does not improve As removal by water leaching.

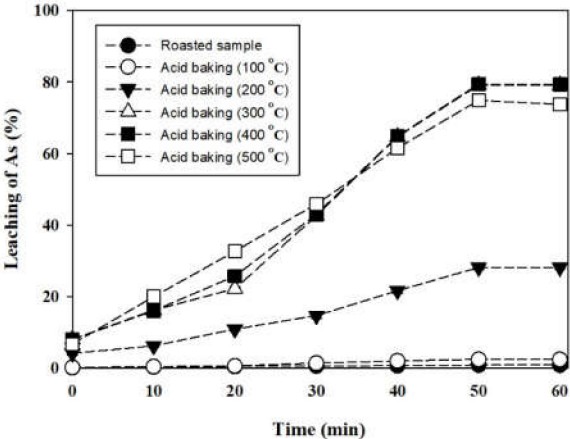

**Figure 5.** Leaching of As in the hot water leaching as a function of time.

The increased As dissolution, up to 400 °C, is due to the accelerated sulfation of sulfide minerals by sulfate solvent; thus, the valuable metals in the soluble mineral could be easily leached. Equations (5)–(7) are the chemical reaction mechanism of $H_2SO_4$ with the product that can be generated by the reaction of Equations (1)–(4).

$$FeSO_4(Ferrous\ sulfate) + 2H_2SO_4 = Fe_2(SO_4)_3 + SO_2 + 2H_2O \tag{5}$$

$$H_2S(Hydrogensulfide) + H_2SO_4 = S^o + SO_2 + 2H_2O \tag{6}$$

$$S^o(Sulfur) + 2H_2SO_4 = 3SO_2 + 2H_2O \tag{7}$$

The slower arsenic leaching by the further increase in baking temperature up 500 °C meant that dissolution decreased due to the mineral phase change in the oxide minerals [14].

#### 3.2.2. Leaching Valuable Metals

Water leaching for 50 min exhibited the maximum leaching of valuable metals. The concentrations of Fe, Cu, Zn, and Pb in the leaching solution for the baked concentrate are shown in Figure 6. The roasted samples (Figure 6a) at higher temperatures showed higher leaching for Fe, Cu, Zn, and Pb, but the concentrations were much lower than those for the acid baking samples (Figure 6b). This means that acid baking enhances the efficiency of leaching for gold concentrate. The maximum leaching of Fe and Pb (76 mg/L and 32.39 mg/L) was shown in the leaching of baked samples at 300 °C. In the cases of Cu and Zn, the maximum leaching (81.37 mg/L and 81.16 mg/L) occurred from the 400 °C baked sample.

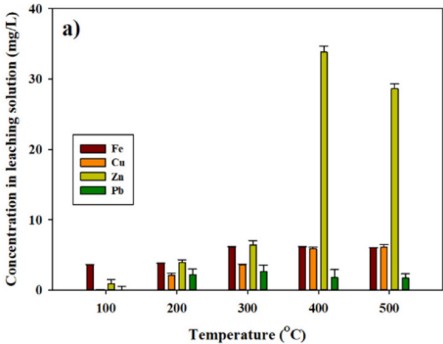
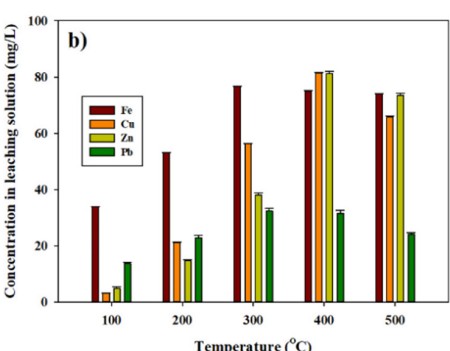

**Figure 6.** Concentration of Fe, Cu, Zn, and Pb in leaching water from the hot water leaching process after (**a**) roasting or (**b**) acid baking at different temperatures from 100 °C to 500 °C.

### 3.2.3. Enrichments in Au and Ag Grade of Solid Residue

The water-leaching experiment was conducted with roasting and baking samples. Since volatile elements such as As were removed in the baking process and soluble metal sulfates were dissolved and removed during the water leaching process, the grades of Au and Ag increased in the solid residue (Figure 7). The temperature at which the maximum enrichment of Au (Figure 7a) grade was shown was 500 °C (4.76%) in the roasting sample, while it was 300 °C (33.03%) in the baking sample. The temperature at which the maximum enrichment of Ag (Figure 7b) grade was shown was 500 °C (0.96%) in the roasting sample, whereas it was 300 °C (15.20%) and 400 °C (15.20%) in the baking sample. Using a roasting-leaching process, the grade of Au and Ag was enrichment by up to 4.76% and 0.96% in the case of the 500 °C roasting sample. However, when acid baking was used, the grades of Au and Ag (33.03% and 15.20%, respectively) could be maximized by 300 °C acid baking.

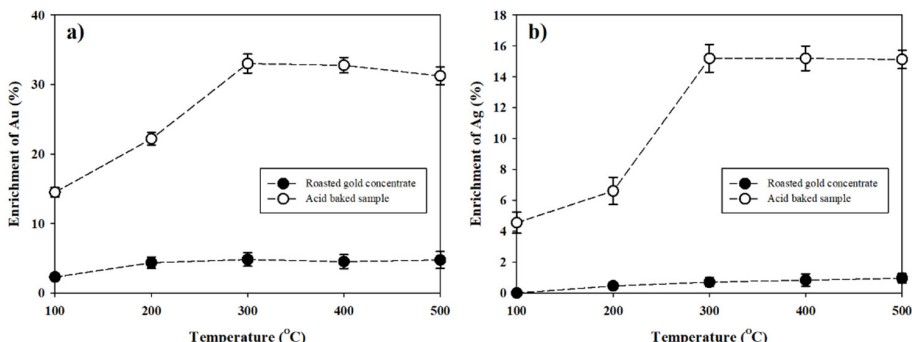

**Figure 7.** Enrichment of (**a**) Au and (**b**) Ag on the solid residue from hot water leaching after acid baking at different temperatures (from 100 °C to 500 °C).

### 3.2.4. Mineral PHASE Change in Solid Residue by Acid Baking-Water Leaching

After the water leaching experiment was complete, insoluble solid residue was filtered and collected. The XRD analysis was conducted to determine the mineralogical properties of the insoluble solid residue (Figure 8). The result showed that quartz and pyrite were revealed as being common to all baking samples in a range of 100–500 °C. Muscovite and elemental sulfur ($S_0$, 3.85 Å and 3.21 Å) were revealed in the 100 °C and 200 °C baking samples, and bassanite ($CaSO_4 \cdot 0.5H_2O$), chalcopyrite, quartz, and pyrite were revealed in the 300 °C baking samples. Bassanite may be formed during the baking process of $H_2SO_4$ or $H_2SO_4$, and is transformed into $SO_2$, which is followed by a reaction between $SO_2$ and CaO to produce $CaSO_4$. After this, it was hydrolyzed with $H_2O$ in a water leaching process, thereby forming $CaSO_4 \cdot 0.5H_2O$ [5]. Galena, hematite, and magnetite were observed in the 400 °C and 500 °C baking samples. However, rhomboclase and anglesite, which were revealed in the baking sample shown in Figure 4, disappeared in the water-leaching solid residue (Figure 8).

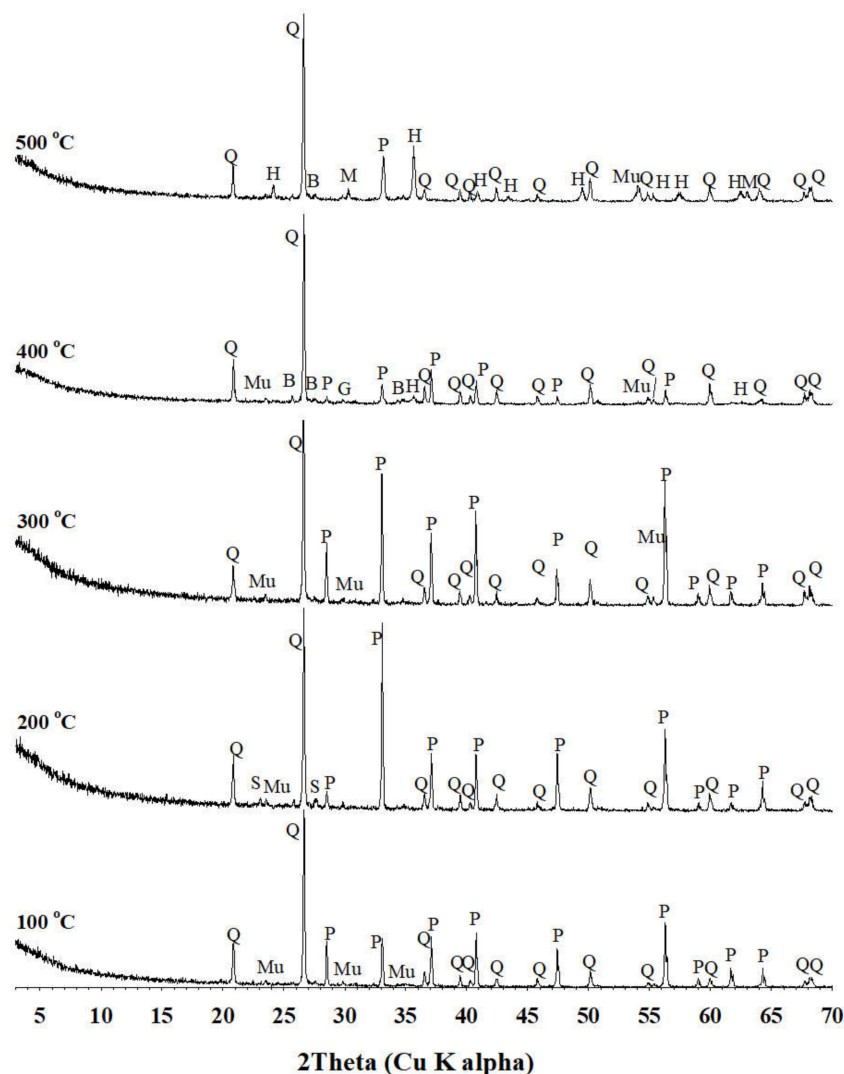

**Figure 8.** XRD patterns of the solid residue after hot water leaching. B, bassanite; G, galena; H, hematite; M, magnetite; Mu, muscovite; P, pyrite; Q, quartz; and S, sulfur.

*3.3. Effect of Operation Conditions for Acid Baking-Water Leaching*

3.3.1. $H_2SO_4$ Dosage in Acid Baking

To determine the effect of $H_2SO_4$ dosage on the performance of the acid baking–water leaching operation, a different volume (1.0, 2.5, 5.0, 6.0, 7.5 mL) of $H_2SO_4$ solution was added for the baking process (for three hours at 400 °C) for the gold concentrate, and water leaching was processed at 70 °C for 50 min for the baked samples. The results of the water-leaching experiment are shown in Figure 9. In Figure 9, the leaching of As increased with an increase in $H_2SO_4$ concentration. It reached a maximum of 79.18 mg/L at 5.0 mL of $H_2SO_4$ concentration. Despite the fact that the $H_2SO_4$ concentrations increased to 6.0 mL (79.44 mg/L) and 7.5 mL (78.18 mg/L), the maximum leaching of As did not increase significantly but showed a nearly constant value. Figure 9 shows the leaching of Fe, Cu, Zn, and Pb. The leaching of Fe also increased with an increase in $H_2SO_4$ concentration, and it reached a maximum of 77.72 mg/L at 5.0 mL of $H_2SO_4$ dosage. Despite the fact that $H_2SO_4$ concentrations increased to 6.0 mL and 7.5 mL, the maximum leaching of Fe decreased compared to that at 5.0 mL. The leaching of Cu increased with an increase in $H_2SO_4$ concentration. A maximum of 74.92 mg/L was revealed in 7.5 mL of $H_2SO_4$ concentration. The maximum leaching of Zn was revealed when the $H_2SO_4$ concentration was 6.0 mL (80.85 mg/L), and the maximum leaching of Pb was also observed when the $H_2SO_4$ concentration was 6.0 mL (50.14 mg/L).

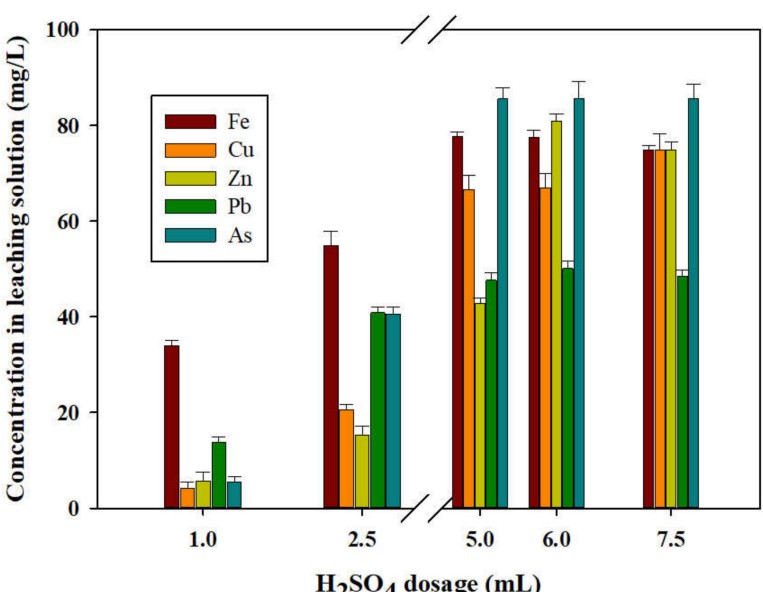

**Figure 9.** Effect of $H_2SO_4$ dosage (1.0, 2.5, 5.0, 6.0, and 7.5 mL) in hot water leaching on leaching rate of valuable metals and As from the acid-baked gold concentrate (at 400 °C, for 3 h).

Arsenic removal and an enrichment in Au/Ag grade were also found in the solid residue. Insoluble solid residue was collected after the 70 °C water-leaching experiment was conducted with the baked samples with $H_2SO_4$ concentration. The aqua regia digestion was conducted with the solid residue, and the contents of As, Au, and Ag between the solid residue and gold concentration sample were compared (Figure 9). In Figure 10, the As removal increased with an increase in $H_2SO_4$ concentration. However, the As removals at 5.0 mL, 6.0 mL, and 7.5 mL concentrations were 88.41%, 88.60%, and 88.49%, respectively. In Figure 10, the Au and Ag grade enrichments reached a maximum level (11.83% and 11.31%, respectively) at 5.0 mL $H_2SO_4$. Since the As removal and Au and Ag enrichments increased insignificantly at 5.0 mL in higher $H_2SO_4$ concentrations, the optimum concentration of $H_2SO_4$ was set to 5.0 mL. Subsequently, $H_2SO_4$ dosage was fixed at 5.0 mL to conduct the next baking experiment at 400 °C according to time.

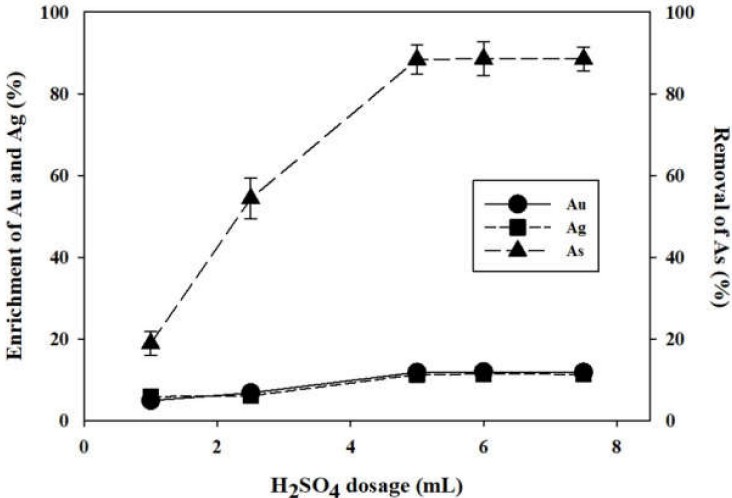

**Figure 10.** The removal of As and the enrichments of Au and Ag on the solid residue from hot water leaching. $H_2SO_4$ dosage (1.0, 2.5, 5.0 6.0, and 7.5 mL); baking temperature, 400 °C; and baking time, 3 h.

3.3.2. Time for Acid Baking

To determine the effect of time in acid baking on the performance of the acid-baking–water leaching operation, acid baking for the gold concentrate was performed at 400 °C for different times, from 3 h to 7 h, and water leaching was processed at 70 °C for 50 min for the baked samples. The results of the water leaching experiment are shown in Figure 11. Figure 11 shows that the As leaching of the roasting sample was 3.64% at three hours of roasting, and 3.32% at seven hours of roasting, which revealed a range of around 3%. However, the As leaching in the baking sample was 81.43% in the four hour baking sample (leaching time = 60 min.), 81.21% in the five hour baking sample (leaching time = 60 min), 80.96% in the six hour baking sample (leaching time = 60 min.), and 79.87% in the seven hour baking sample (leaching time = 60 min.). The maximum As leaching was exhibited in the four hour baking sample. Although baking was increased to four hours or longer, the As leaching did not increase to more than 81%. A difference in the maximum As leaching between four hour roasting and baking samples was more than 21-fold. The baking time when Fe was leached the most was four hours, and its leaching was 80.02%. All of the baking times when Cu, Zn, and Pb were leached, the most were five hours, and their leaching were 70.10%, 81.93%, and 50.48%, respectively. Differences in the maximum leaching in Fe, Cu, Zn, and Pb between roasting and baking times were more than 17-, 10-, 14-, and 13-fold, respectively.

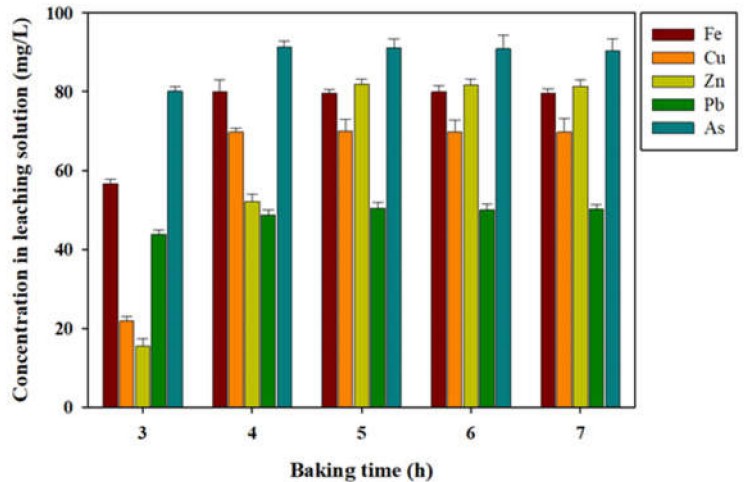

**Figure 11.** Leaching valuable metals and As on the hot water leaching as a function of time. Condition, $H_2SO_4$; concentration, 1.0 mL; and baking temperature, 400 °C.

Arsenic removal and enrichments in Au/Ag grade were also found in the solid residue. The aqua regia digestion was conducted with the insoluble solid residue to analyze the contents of As, Au, and Ag. Figure 12, showed that As removals were 92.00%, 92.03%, 92.00%, 91.91%, and 91.89% at three, four, five, six, and seven hour baking samples, respectively. The Au grade enrichment showed a value of 15.51% (four hours) from the minimum to 16.59% (five hour) to the maximum after baking from three hours to seven hours. The Ag grade enrichment showed a value of 11.66% (four hours) from the minimum to 11.86% (six hours) at the maximum after baking from three hours to seven hours. Thus, more than 91% of the As content of the 2456.1 mg/kg sample was removed when $H_2SO_4$ was added to the gold concentrate sample and baked for three hours at 400 °C. The grades of Au and Ag also enrichments by 15% and 11% or higher, respectively.

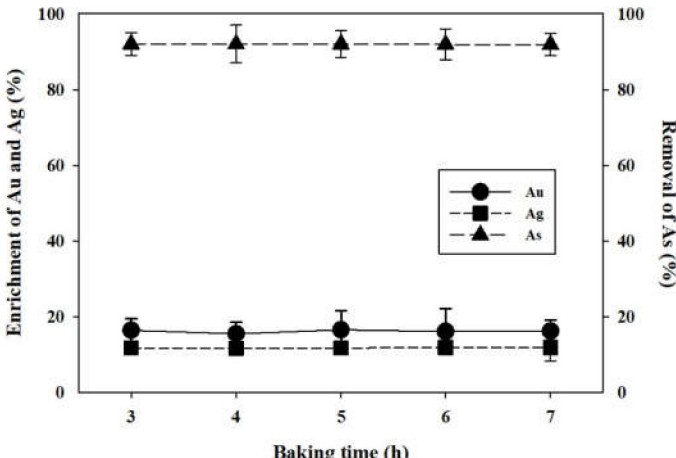

**Figure 12.** The removal of As and the enrichments of Au and Ag on the solid residue from hot water leaching. Conditions: baking temperature, 400 °C; $H_2SO_4$ concentration, 1.0 mL; and baking time, 3, 4, 5, 6, 7 h.

## 4. Discussion

Acid baking for gold concentration can be expected to enrich Au/Ag and remove arsenic. The contents of Au and Ag increased when the gold concentrate was baked with $H_2SO_4$. Furthermore, the content of Au increased when $H_2SO_4$ concentration increased (Figure 5). However, the Au increase rate did not exceed 9% in the roasting and baking experiments. The reason for this was because the heating, roasting, and baking were conducted at relatively low temperatures, such as 400 °C and 500 °C. When roasting for a long time at 750 °C or higher, where pyrite and arsenopyrite are thermally decomposed, an Au enrichment of 9% or more was obtained [7,25,27].

Arsenic was partially removed, even by a simple roasting process, and more removal was possible through the baking process using $H_2SO_4$. When calcination is carried out by increasing the $H_2SO_4$ concentration, the removal of arsenic (volatilization) may increase; however, it was confirmed that the As removal did not exceed 10% under the conditions of this study. The reason for this is that the As removal in the roasting and baking experiments was at 400 °C or 500 °C, i.e., the roasting and baking temperatures were relatively low [28].

In previous studies, it was shown that over 95% of As was removed when being roasted for a long time at 700 °C or higher, i.e., where arsenopyrite is thermally decomposed [26,29]. The water (70 °C) leaching after acid baking showed that As was leached 60 times more in the baking sample (78.20%) than that in the roasting sample (1.3%). In addition, 81% of As was leached when the $H_2SO_4$ concentration was increased and roasting time was increased. This leaching was obtained when As was dissolved with water. However, if the gold concentrate is simply roasted at a temperature of 700 °C or higher, almost 100% of As can be removed; that said, arsenic is released as $As_2O_3$ and becomes an air pollutant [7,30–32].

Furthermore, when the gold concentrate was baked with 5.0 mL $H_2SO_4$ and baking was conducted with an increase in $H_2SO_4$ concentration, 76–77% of Fe, 74–81% of Cu, 80–81% of Zn, and 32–50% of Pb were leached in the water leaching process. On the other hand, if the gold concentrate was roasted at 750–800 °C for the purpose of gold recovery, and cyanide or non-cyanide solvent was applied, a higher gold recovery would be obtained. Although the above roasting method can extract gold with a high recovery, it can also transform valuable metals such as Fe, Cu, Zn, and Pb contained in the gold concentrate, thereby losing them as tailings. Fe, Cu, or Zn contained in the gold concentrate is transformed to a metal sulfate when the gold concentrate is baked with $H_2SO_4$ and then leached into water; the gold contained in the gold concentrate is not dissolved in water since it does not form sulfate and does not react, even when baking with $H_2SO_4$. Thus, gold remains in the insoluble solid residue. Since most valuable metals in the gold

concentrate are transformed to soluble metal sulfate during the baking process, they are dissolved and removed in water leaching. Because of this, the gold content naturally increases in the solid residue. Thus, the enrichment in Au grade in the solid residue was verified after water leaching. Although Au grade was revealed as being up to 4% in the roasting sample (500 °C), Au grade in the baking sample was observed as being up to 33% (300 °C). The reason for the enrichment in Au grade in the solid residue was because metals contained in the gold concentrate were transformed into metal sulfate that could be easily dissolved in water during the baking process with $H_2SO_4$. The XRD result verified that rhomboclase ($FeH[SO_4]_2 \cdot 4H_2O$) and mikasite ($Fe_2[SO_4]_3$), which are metal sulfates, were produced during the baking process. Although anglesite ($PbSO_4$) was produced by baking, which was verified through XRD, the existence of $CuSO_4$ or $ZnSO_4$ was not verified. However, $CuSO_4$ and $ZnSO_4$ seemed to be produced during the baking process because Cu and Zn revealed a very high leaching rate in water leaching. In the XRD analysis, the existence of rhomboclase and mikasite can be verified, whereas the production of copper sulfate or zinc sulfate cannot be verified. The reason for this was because the content of Fe (86,793.1 mg/kg) was very high in the gold concentrate, whereas the contents of Cu (1899.4 mg/kg) and Zn (2709.6 mg/k) were very low. Although copper sulfate or zinc sulfate were generated during the baking process, their amounts were very small and could not be detected by the XRD analysis [33].

## 5. Conclusions

$H_2SO_4$ was added to gold concentrate, and baking experiments were conducted at a relatively low temperature in an electric furnace. As a result, rhomboclase and mikasite were produced in the soluble baking sample, which was verified through XRD. The leaching experiment was conducted with roasted and acid-baked samples at 70 °C water, and the results showed that As, Fe, Cu, Zn, and Pb were dissolved with very high leaching rates in the acid-baked samples compared to those in the roasted samples. Various experiments with regard to baking effect, $H_2SO_4$ concentration effect, and baking time effect were conducted; the soluble mineral results showed As values of more than 60-fold were leached in the baking sample compared to the roasting sample, and values 17-, 10-, 14-, and 13-fold greater for Fe, Cu, Zn, and Pb, respectively, were leached. Gold grade was enrichment by 33% or more in the water-leached solid residue. If the gold concentrate had been roasted, As would have been released into the atmosphere to become an air pollutant, while valuable metals such as Fe, Cu, Zn, and Pb would have been transformed into oxides, thereby losing them as tailings. However, the result of baking with $H_2SO_4$ showed that As was recovered with a high leaching rate in the water-leaching process, thereby avoiding As becoming an air pollutant while valuable metals such as Fe, Cu, Zn, and Pb were recovered from the water leaching to prevent a loss of them as oxides.

**Author Contributions:** B.K. performed writing—original draft and funding acquisition. C.P. performed conceptualization and validation. K.C. performed visualization. J.K. performed writing-review and editing. N.C. performed supervision. S.L. performed writing—review and editing, and funding acquisition. All authors have read and agreed to the published version of the manuscript.

**Funding:** This work was supported by Korea Environment Industry & Technolgy Institute (KEITI) through Subsurface Environmental Pollution Risk Management Technology Development Project, funded by Korea Ministry of Environmenta(MOE)(ARQ202101728001), and by the Institute for Korea Spent Nuclear Fuel (iKSNF) and National Research Foundation of Korea(NRF) grant funded by the Korea government (Ministry of Science and ICT, MSIT) (2021M2E1A1085202).

**Conflicts of Interest:** The authors declare no conflict of interest.

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
