# Peer review of "Sulfuric Acid Baking—Water Leaching for Gold Enrichment and Arsenic Removal from Gold Concentrate"

_minerals, doi:10.3390/min11121332_

Round 1

Reviewer 1 Report

The article presents the results of experimental studies to improve the methods of extracting gold and other valuable metals from sulfide minerals containing invisible gold. In addition, it was studied ways to improve the quality of gold with invisible gold for leaching valuable metals and removing penalty elements using the method of acid leaching in hot water. The article presents the results of studying the mineral composition of the gold concentrate sample and products of their processing by modern mineralogical methods.The data interpretations seem to have been done well. So, the paper adds valuable new information and can recommend the paper for publication in Minerals but with some revisions. These revisions relate to the Figures, their quality should be improved.

Author Response

  1. These revisions relate to the Figures, their quality should be improved.

Answer) At first, we are appreciated for your review and the kind comments. All figures have been redone to improve quality and readability. And the manuscript has undergone English language editing by MDPI, again.

Reviewer 2 Report

This manuscript reports on work that is relevant to the field of Minerals, in its current state the manuscript cannot be published for many different reasons. The style is readable, despite the quirks of language. The coverage and content of the manuscript are balanced, although some parts need more clear description. I would recommend that the manuscript should be accepted after major revision.

Methodology part is not clear, how many samples were used for the experiments? Statistics are missing. What about Anova, or at least SD?

Abbreviations are not explained when used for the first time. I would suggest doing the list of abbreviations used.

Generally capute to Figres do not match. For example:  Fig. 9 axis OX, the value of temperature is not correct. It is 1-8 oC, which is not true. Here also the description does not match. 

Fig. 10 is signed as "Leaching rate of As on the hot water-leaching as a function of time." but this shows concentration and baking time for As, Fe, Cu, Zn , Pb. 

Units should be presented in consistent way, for example mL or ml. Please do not mix.

 On p. 3 Authors explained what is XRD but do not give an explanation of what is XRF.

Fig. 2, 4 and 6, Authors used "rate" - here is not a rate (rate is when you have per time unit)

Fig. 8 What is the concentration means on OY? Please specify.

Fig. 11 should be combined in one, not a, b and c.

Fig. 9 and 11. It is not a removal rate, because "the rate" should be per time unit. Here it is only removal.

Author Response

  1. The style is readable, despite the quirks of language. The coverage and content of the manuscript are balanced, although some parts need more clear description. I would recommend that the manuscript should be accepted after major revision.

answer) We revised overall manuscript again. And the revised manuscript has been undergone English language editing by MDPI.

  1. Methodology part is not clear, how many samples were used for the experiments? Statistics are missing. What about Anova, or at least SD? 

answer) Most samples were analyzed in duplicate, and an error bar was added to the figure.

  1. Abbreviations are not explained when used for the first time. I would suggest doing the list of abbreviations used.

answer) The abrreviations such as XRD and AAS are explained when used for the fisrt time (Line 121 - 122). 

  1. Generally caption to Figures do not match. For example:  Fig. 9 axis OX, the value of temperature is not correct. It is 1-8 oC, which is not true. Here also the description does not match. 

answer) We corrected all figures and the descriptions during the overall revision.  

  1. Fig. 10 is signed as "Leaching rate of As on the hot water-leaching as a function of time." but this shows concentration and baking time for As, Fe, Cu, Zn , Pb.

answer) We prepared the figure again. And the caption was corrected as "Leaching rate of valuable metals and As ..."

  1. Units should be presented in consistent way, for example mL or ml. Please do not mix.

answer) The units of volume and concentration corrected with mL and mg/L, consistently. 

  1. On p. 3 Authors explained what is XRD but do not give an explanation of what is XRF.

answer) There was a mistake to explain the chemical analysis method. XRF was corrected with AAS in the manuscript. 

  1. Fig. 2, 4 and 6, Authors used "rate" - here is not a rate (rate is when you have per time unit)

answer) As the reviewer said, "rate" is often used as a concept related to time. However, it can be used instead of terms such as "ratio, percentage" in the dictionary meaning. Therefore, I think there is no problem in the use of the word "rate" in this paper.

  1. Fig. 8 What is the concentration means on OY? Please specify.

answer) We corrected figures according to reviewer comments.  

  1. Fig. 11 should be combined in one, not a, b and c. 

answer) We corrected figures according to reviewer comments.

  1. Fig. 9 and 11. It is not a removal rate, because "the rate" should be per time unit. Here it is only removal.

answer) As the reviewer said, "rate" is often used as a concept related to time. However, it can be used instead of terms such as "ratio, percentage" in the dictionary meaning. Therefore, I think there is no problem in the use of the word "rate" in this paper.

Reviewer 3 Report

Manuscript ID: minerals-1441175

Title: Sulfuric Acid Bake - Water Leaching for Gold enrichment and Arsenic Removal from Gold Concentrate

Authors: Bong-Ju Kim et al.

Title. Authors must write «baking» or «digestion», not bake.

Line 41, 66, 70, 80, 105. Change penalty elements to impurities.

Line 82-89. Authors must separate the references to 1-2 references for one sentence or add references for each metal Cu [], Fe [], Zn [], etc.

Line 95. Re-write “molten metal”, it is not correct words.

Table 1. Main elements must write by wt.%. What is the LOI (loss on ignition at 1000 °C) content in gold concentrate?

Section 2.2.1 What is the H2SO4 concentration?

Line 148-149. Why so big liquid to solid ratio = 50?

Figure 2. Title was incorrect.

Figure 3. Resolution very bad. Improve it.

Section 3.1.2 Add chemical reactions of phase transformations of mineral phases of gold concentrate by baking process.

Figure 4. Authors must add error bars. Why at 500 °C the leaching rate of As was decreased?

Figure 5, 8, 10. Authors must increase the font size on the figure and add error bars.

Section 3.2.4 Add chemical reactions of water leaching process.

Figure 7. Resolution very bad. Improve it.

Figure 9. The topic of X-axis was incorrect.

  • Authors must add the flowsheet of the acid baking - water leaching process.
  • What is the chemical and mineral compositions and of solid residue after water leaching?

References are very old. Add new links (10-15) from 2019-2021.

Technical errors:

Line 33. Change the ㎛ to microns.

Line 120. Add chemical formula of muscovite.

Author Response

At first, we are appreciated for your review and the kind comments. All figures have been redone to improve quality and readability. And the manuscript was edited by a native-speaking reviewer again. The response to the reviewers comments are followed point-by-point. 

1. Authors must write «baking» or «digestion», not bake.

Answer) We used «baking» in our manuscript as your comment.

2. Line 41, 66, 70, 80, 105.Change penalty elements to impurities.

Answer) We replace [penalty elements] in our manuscript to [impurities] as your comment.

3. Line 82-89.Authors must separate the references to 1-2 references for one sentence or add references for each metal Cu [], Fe [], Zn [], etc.

Answer) We separate and relocate the citations as your comments. (Line 86-87)

4. Line 95.Re-write “molten metal”, it is not correct words.

Answer) Thanks for your comment. It corrected with “metal leaching”. (Line 97)

5. Table 1.Main elements must write by wt.%. What is the LOI (loss on ignition at 1000 °C) content in gold concentrate?

Answer) There was an error to describe analysis method of the chemical composition. The chemical composition was analysed usign AAS after aqua regia digestion. So it is not necessary to correct the units and LOI in Table 1. (Line 121-122)

6. Section 2.2.1What is the H2SO4 concentration?

Answer) We added the concentration of H2SO4 (1.0 M) in Line 139.

7. Line 148-149.Why so big liquid to solid ratio = 50?

Answer) In this study, experiments were conducted in a high-liquid ratio environment sufficient for the purpose of confirming the effect of the new process. Previous studies have also conducted studies under similar high-liquid ratio conditions. It is thought that it is necessary to realize the high-liquid ratio condition for future process efficiency evaluation and optimization. Thanks for the detailed advice.

8. Figure 2.Title was incorrect.

Answer) We corrected the caption (Figure 3).

9. Figure 3.Resolution very bad. Improve it.

Answer) We prepared the Figure again to improve the quality (Figure 4).

10. Section 3.1.2Add chemical reactions of phase transformations of mineral phases of gold concentrate by baking process.

Answer) We added the reaction equation as your comments. (Line 221-227)

11. Figure 4.Authors must add error bars. Why at 500 °C the leaching rate of As was decreased?

Answer) The test in Figure 4 was not conducted multiplicately. Unfortunately, we could not add the error bars.

12. Figure 5, 8, 10.Authors must increase the font size on the figure and add error bars.

Answer) We correct the figures as your comments.

13. Section 3.2.4Add chemical reactions of water leaching process.

Answer) we added the chemical reactions in Line 243-245

14. Figure 7.Resolution very bad. Improve it.

 Answer) We prepare the figure again to improve the quality.

15. Figure 9.The topic of X-axis was incorrect.

Answer) We correct the figure agian (Figure 10).  

16. Authors must add the flowsheet of the acid baking - water leaching process.

Answer) We added the flowsheed fo the acid baking – water leaching process in Figure 2.

17. What is the chemical and mineral compositions of solid residue after water leaching?

Answer) The chemical composition of solid residue was analyzed usign AAS after aqua regia digestion, and described focused on Au and Ag in Figure 10. The mineral composition of solid residur was analyzed using XRD as shown in Figure 8 and described in the manuscript.

18. References are very old. Add new links (10-15) from 2019-2021.

Answer) We added new references.

19. Line 33.Change the ㎛ to microns.

Answer) We corrected it as your comments.

20. Line 120.Add chemical formula of muscovite.

Answer) We added chemical formula of muscovite as your comments.

Reviewer 4 Report

Dear Authors, you should read the manuscript carefully again to remove any imperfections in the form of: sticky words, no punctuation marks, correct language used.
The charts in figures 2, 8, 9, 10, 11 are unfortunately not readable, so they should be corrected.
In my opinion, the applied research methodology is correct and appropriate. However, there is no indication of the novelty or originality of the research carried out. Especially when compared to the results of other scientists. To increase the value of the script, complete the information provided
Kind regards

Reviewer

Author Response

  1. Dear Authors, you should read the manuscript carefully again to remove any imperfections in the form of: sticky words, no punctuation marks, correct language used.

Answer) At first, we are appreciated for your review and the comments. As your comment. English proofreading was performed to compensate for the incompleteness of the expression in this paper.

  1. The charts in figures 2, 8, 9, 10, 11 are unfortunately not readable, so they should be corrected.

Answer) We corrected all the charts again to improve the readability.

  1. There is no indication of the novelty or originality of the research carried out. Especially when compared to the results of other scientists.

Answer) We think that our paper deals to a novel approach to enrich low-grade gold ores in order to reduce the air pollution due to As evaporation and recover as much as possible other valuables like Fe, Cu, Zn and Ag. This  method could be compared with the traditional roasting procedure and the benefits.

  1. To increase the value of the script, complete the information provided.

answer) In order to compensate for the incompleteness of this paper, we reviewed overall manuscript and added discussion based on the references. The important changes in our manuscript were indicated in red letters.

Reviewer 5 Report

Dear Authors,

Your paper “Sulfuric Acid Bake - Water Leaching for Gold enrichment and Arsenic Removal from Gold Concentrate” deals to a novel approach to enrich low-grade gold ores in order to reduce the air pollution due to As evaporation and recover as much as possible other valuables like Fe, Cu, Zn and Ag. The new method was compared with the traditional roasting procedure and the benefits are highlighted.

Even the topic is really worth of investigation, the manuscript suffers mainly of a very poor English style, that often not allow a correct understanding of the main findings and explanations.

In addition, also the discussion I poor, even to a lesser extent than English style. Generally, literature comparison is modest and often the involved mechanisms about Au and Ag enrichment, as well as As removal are not fully described neither discussed. For these reasons, I highly recommend to review the paper with the help of a mother tongue English editor as well as to improve the discussion adding explanations from a physic-chemical point of view.

Here the Authors can find a detailed list of other changes/corrections/suggestions to further improve their work.

General remarks

  1. The English style is very poor. Some passages in the text are completely unclear. Also, the grammar need a careful revision by a mother tongue reviewer
  2. The results are poorly discussed, especially with similar references. My warm suggestion is to improve the discussion by adding more references to improve the comparison between the Authors’ work and the available literature

Abstract

  1. Line 23. Please check the sentence: “hot water leaching of the roast and baked samples, As the contents leached were 60 times more…”. It seems not correct from gramma point of view

Results and discussion

  1. Figure 2: the caption does not correspond to the figure. Fig2a is related to As removal while Fig2b and c are related to Au and Ag enhancing.
  2. Figure 1, 3 and 7: please provide high-quality XRD pictures. The current figures 1, 3 and 7 are hard to be read.
  3. Line 223-225: “The lower As leaching rate from the roasted sample at 500 ℃ than that at 400 ℃ indicates that baking at higher temperature than 400°C does not help to improve the As removal by water leaching…”. Why? Please add an explanation, maybe supported by literature
  4. Figure 11: Why there was a drop in Au and Ag (Even marginally) enhancing at 4 hours baking time?
  5. Line 400-402: “Cu and Zn were very low. That is, although copper sulfate or zinc sulfate were generated during the baking process, their amounts were very small, which could not be detected by the XRD analysis [27]”. Why did the Authors not performed an SEM analysis to visualize and confirm the formation of Cu and Zn sulfate? By and EDX analysis it would be possible to clearly identify such phases if formed.

Best Regards

Author Response

At first, we are appreciated for your review and the kind comments. Your comments was very helpful to improve our research. Response for your review comments are as follows. 

  1. The English style is very poor. Some passages in the text are completely unclear. Also, the grammar need a careful revision by a mother tongue reviewer.

Answer) As your comment, our manuscript was revised carefullly, and undergone English language editing by MDPI to make up for the incomplete expression in this paper.

  1. The results are poorly discussed, especially with similar references. My warm suggestion is to improve the discussion by adding more references to improve the comparison between the Authors’ work and the available literature

Answer) In order to compensate for the incompleteness of this paper, we reviewed overall manuscript and added discussion based on the references. Importance changes in our manuscript were indicated in red letters.

  1. Abstract Line 23. Please check the sentence: “hot water leaching of the roast and baked samples, As the contents leached were 60 times more…”. It seems not correct from gramma point of view

Answer) We corrected the grammatical error. Thanks for your kind comment

  1. Results and discussion Figure 2: the caption does not correspond to the figure. Fig2a is related to As removal while Fig2b and c are related to Au and Ag enhancing.

Answer) We corrected figure 2 and its caption.

  1. Figure 1, 3 and 7: please provide high-quality XRD pictures. The current figures 1, 3 and 7 are hard to be read.

Answer) We prepare the figure again with higher-resolution.

  1. Line 223-225: “The lower As leaching rate from the roasted sample at 500 ℃ than that at 400 ℃ indicates that baking at higher temperature than 400°C does not help to improve the As removal by water leaching…”. Why? Please add an explanation, maybe supported by literature

Answer) We explain the reason of slower arsenic leaching in Line 246-247 based on the literature.

  1. Figure 11: Why there was a drop in Au and Ag (Even marginally) enhancing at 4 hours baking time?

Answer) We plotted Figure 12 again. We considered that the reduction in Au and Ag enhancement at 4 hr of baking was negligible. We could not found any reasons to explain this small difference. 

  1. Line 400-402: “Cu and Zn were very low. That is, although copper sulfate or zinc sulfate were generated during the baking process, their amounts were very small, which could not be detected by the XRD analysis [27]”. Why did the Authors not performed an SEM analysis to visualize and confirm the formation of Cu and Zn sulfate? By and EDX analysis it would be possible to clearly identify such phases if formed.

answer) Thanks for your comment. As your comment, the SEM EDX could have helped to identify the phase transformation. Unfortunately, we could not conduct EDX analysis. In further study, we’ll keep in mind your valuable comments.

Round 2

Reviewer 3 Report

The authors answered in detail all the questions. I agree with the answers of the authors.

However, there are a number of comments on the design of the article:
1) Figures 2, 3, 5, 6, 7 The authors must write degrees correctly - °C
2) Figures 2, 11, 12 - Use h for time, not hr
3) Figures 6, 9, 11 - it is necessary to use different colors for columns - blue, green orange, red, and so on. It is very difficult to distinguish gray colors.

Author Response

1) Figures 2, 3, 5, 6, 7 The authors must write degrees correctly - °C

Answer) We corrected figures according to reviewer comments.

2) Figures 2, 11, 12 - Use h for time, not hr

Answer) We corrected figures according to reviewer comments.

3) Figures 6, 9, 11 - it is necessary to use different colors for columns - blue, green orange, red, and so on. It is very difficult to distinguish gray colors.

Answer) We corrected figures according to reviewer comments.

Reviewer 5 Report

Dear Authors,

thank you for submitting a revised version of your paper

After a careful reading, my feeling is that the most of the reviewers' requests were well addressed.

I still have some  minor revision pose to the Authors attention:

1) p.5, line 182-191. From your description it is not clear where the As goes during acid baking. Does As remain in the solution or volatilize like during roasting? The same question about Au and Ag. Are they concentrating in the solution or in the solid residue?

2) p.5, line 194-208. What about the roasted samples? Why the Authors did not reported the XRD of roasted samples at the different temperature? It would be interesting to make a direct comparison between acid baking and roasting at the same temperature in terms of crystallographic modification.

3) p.7, line 229. In the experimental procedure it was stated that 2.0 grams of material undergo the leaching. Howevere, in the section 3.2.1, it was reported 1.0 gram. Which is the right amount used for leaching?

4) p.8, line 261-273 + p.9, line 277-290. Au and Ag in which form are present within the solid residue? Metallic, oxide, sulfide? Why there is no phases associated to Au and Ag in the XRD reported in Figure 8? From the chemical analysis, the concentration of Au and Ag are not negligible to be not detected by XRD.

5) general remak: please, try to better distinguish the samples from acid-baking from those from rosting. For instance, use acid-baking for this group and roasted for the samples being heated without acid.

Best Regards

Author Response

1) p.5, line 182-191. From your description it is not clear where the As goes during acid baking. Does As remain in the solution or volatilize like during roasting? The same question about Au and Ag. Are they concentrating in the solution or in the solid residue?

Answer) We are proud of your constructive comments. In the case of arsenic, some volatilization occurs and the solution is concentrated, and in the case of gold/silver, it is concentrated in the residue.

p.5, Line: 188-191

“As a result of acid baking, the material flow was volatilized in small amounts in the case of As(As2O3). In the acid baking experiment, arsenic exists in the form of arse-nic acid in the solution, and in the case of Au and Ag, it exists in the form of AuO and Ag2O in the residue”

2) p.5, line 194-208. What about the roasted samples? Why the Authors did not reported the XRD of roasted samples at the different temperature? It would be interesting to make a direct comparison between acid baking and roasting at the same temperature in terms of crystallographic modification.

Answer) We are proud of your constructive comments.

Thank you very much for asking precisely what we are considering.

The roasting sample was XRD-analyzed, but not indicated in the text.

The reason for this is that although the phase change of the Fe mineral in the raw material was made, it was difficult to see any other changes other than the change of hematite → magnetite in Pyrite. Therefore, in this paper, we tried to focus on improving the quality of Au and Ag due to the phase change by acid baking.

3) p.7, line 229. In the experimental procedure it was stated that 2.0 grams of material undergo the leaching. Howevere, in the section 3.2.1, it was reported 1.0 gram. Which is the right amount used for leaching?

Answer) Thank you for your kind comments. We corrected in the text.

p.7, Line : 233, “with 2.0 g”

4) p.8, line 261-273 + p.9, line 277-290. Au and Ag in which form are present within the solid residue? Metallic, oxide, sulfide? Why there is no phases associated to Au and Ag in the XRD reported in Figure 8? From the chemical analysis, the concentration of Au and Ag are not negligible to be not detected by XRD.

Answer) In the case of acid baking experiment, Au and Ag exist in the form of AuO and Ag2O in the residue. The existence of AuO(oxide) and Ag2O(oxide) can be observed by XRD, but the presence of peaks could not be confirmed because the content of Fe mineral and quartz was relatively high. In addition, it was difficult to observe since the increased amounts of Au and Ag were below the detection limit value through general XRD analysis.

With reference to these points, we will check the opinions of the reviewers in future research. Thank you for your kind comments.

5) general remak: please, try to better distinguish the samples from acid-baking from those from rosting. For instance, use acid-baking for this group and roasted for the samples being heated without acid.

Answer) Thank you for your kind comments. It has been rewritten separately in the text.

p.5, Line : 182

“In the roasting process (roasted for the samples being heated without acid(H2SO4))”